# Black-box Off-policy Estimation for Infinite-Horizon Reinforcement Learning

**Ali Mousavi**[*]  **Lihong Li**[*]  **Qiang Liu**[†]  **Dengyong Zhou**[*]

[*]Google Research
{alimous,lihong,dennyzhou}@google.com

[†]University of Texas, Austin
lqiang@cs.utexas.edu

## Abstract

Off-policy estimation for long-horizon problems is important in many real-life applications such as healthcare and robotics, where high-fidelity simulators may not be available and on-policy evaluation is expensive or impossible. Recently, Liu et al. (2018) proposed an approach that avoids the *curse of horizon* suffered by typical importance-sampling-based methods. While showing promising results, this approach is limited in practice as it requires data be drawn from the *stationary distribution* of a *known* behavior policy. In this work, we propose a novel approach that eliminates such limitations. In particular, we formulate the problem as solving for the fixed point of a certain operator. Using tools from Reproducing Kernel Hilbert Spaces (RKHSs), we develop a new estimator that computes importance ratios of stationary distributions, without knowledge of how the off-policy data are collected. We analyze its asymptotic consistency and finite-sample generalization. Experiments on benchmarks verify the effectiveness of our approach.

## 1 Introduction

As reinforcement learning (RL) is increasingly applied to crucial real-life problems like robotics, recommendation and conversation systems, off-policy estimation becomes even more critical. The task here is to estimate the average long-term reward of a target policy, given historical data collected by (possibly unknown) behavior policies. Since the reward and next state depend on what action the policy chooses, simply averaging rewards in off-policy data does not estimate the target policy's long-term reward. Instead, proper correction must be made to remove the bias in data distribution.

One approach is to build a simulator that mimics the reward and next-state transitions in the real world, and then evaluate the target policy against the simulator (e.g., Fonteneau et al., 2013; Ie et al., 2019). While the idea is natural, building a high-fidelity simulator could be extensively challenging in numerous domains, such as those that involve human interactions. Another approach is to use propensity scores as importance weights, so that we could use the weighted average of rewards in off-policy data as a suitable estimate of the average reward of the target policy. The latter approach is more robust, as it does not require modeling assumptions about the real world's dynamics. It often finds more success in short-horizon problems like contextual bandits, but its variance often grows exponentially in the horizon, a phenomenon known as "the curse of horizon" (Liu et al., 2018).

To address this challenge, Liu et al. (2018) proposed to solve an optimization problem of a minimax nature, whose solution directly estimates the desired propensity score of states *under the stationary distribution*, avoiding an explicit dependence on horizon. While their method is shown to give more accurate predictions than previous algorithms, it is limited in several important ways:

- The method requires that data be collected by a known behavior policy. In practice, however, such data are often collected over an extended period of time by multiple, unknown behavior policies. For example, observational healthcare data typically contain patient records, whose treatments were provided by different doctors in multiple hospitals, each following potentially different procedures that are not always possible to specify explicitly.

- The method requires that the off-policy data reach the stationary distribution of the behavior policy. In reality, it may take a very long time for a trajectory to reach the stationary distribution, which may be impractical due to various reasons like costs and missing data.

In this paper, we introduce a novel approach for the off-policy estimation problem that overcome these drawbacks. The main contributions of our work are three-fold:

- We formulate the off-policy estimation problem into one of solving for the fixed point of an operator. Different from the related, and similar, Bellman operator that goes forward in time, this operator is backward in time.
- We develop a new algorithm, which does not have the aforementioned limitations of Liu et al. (2018), and analyze its generalization bounds. Specifically, the algorithm does not require that the off-policy data come from the stationary distribution, or that the behavior policy be known.
- We empirically demonstrate the effectiveness of our method on several classic control benchmarks. In particular, we show that, unlike Liu et al. (2018), our method is effective even if the off-policy data has not reached the stationary distribution.

In the next section, we give a brief overview of recent and related works. We then move to describing the problem setting that we have used in the course of the paper and our off-policy estimation approach. Finally, we present several experimental results to show the effectiveness of our method.

**Notation.** In the following, we use $\Delta(\mathcal{X})$ to denote the set of distributions over a set $\mathcal{X}$. The $\ell_2$ norm of vector $x$ is $\|x\|$. Given a real-valued function $f$ defined on some set $\mathcal{X}$, let $\|f\|_2 := \sqrt{\int_{\mathcal{X}} f(x)^2 dx}$. Finally, we denote by $[n]$ the set $\{1, 2, \ldots, n\}$, and $\mathbf{1}\{A\}$ the indicator function.

## 2 RELATED WORKS

Our work focuses on estimating a scalar (average long-term reward) that summarizes the quality of a policy and has extensive applications in practice. This is different from value function or policy learning from off-policy data (e.g., Precup et al., 2001; Maei et al., 2010; Sutton et al., 2016; Munos et al., 2016; Metelli et al., 2018), where the major goal is to ensure stability and convergence. Yet, these two problems share numerous core techniques, such as importance reweighting and doubly robustness. Off-policy estimation and evaluation can also be used as a component for policy optimization (e.g., Jiang & Li, 2016; Gelada & Bellemare, 2019; Liu et al., 2019; Zhang et al., 2019).

Importance reweighting, or inverse propensity scoring, has been used for off-policy RL (e.g., Precup et al., 2001; Murphy et al., 2001; Li et al., 2015; Munos et al., 2016; Hanna et al., 2017; Xie et al., 2019). Its accuracy can be improved by various techniques (Jiang & Li, 2016; Thomas & Brunskill, 2016; Guo et al., 2017; Farajtabar et al., 2018). However, these methods typically have a variance that grows exponentially with the horizon, limiting their application to mostly short-horizon problems like contextual bandits (Dudík et al., 2011; Bottou et al., 2013).

There have been recent efforts to avoid the exponential blow-up of variance in basic inverse propensity scoring. A few authors explored the alternative to estimate the propensity score of a state's *stationary distribution* (Liu et al., 2018; Gelada & Bellemare, 2019), when behavior policies are known. Later, Nachum et al. (2019) extended this idea to situations with *unknown* behavior policies. However, their approach only works for the discounted reward criterion. In contrast, our work considers the more general and challenging *undiscounted* criterion. In the next section, we briefly mention the setting under which we study this problem and then present our *black-box* off-policy estimator.

Our black-box estimator is inspired by previous work for black-box importance sampling (Liu & Lee, 2017). Interestingly, the authors show that it is beneficial to estimate propensity scores from data without using knowledge of the behavior distribution (called proposal distribution in that paper), *even if* it is available; see also Henmi et al. (2007) for related arguments. Similar benefits may exist for our black-box off-policy estimator developed here, although a systematic study is outside the scope of this paper.

## 3 PROBLEM SETTING

Consider a Markov decision process (MDP) (Puterman, 1994) $M = \langle \mathcal{S}, \mathcal{A}, P, R, p_0, \gamma \rangle$, where $\mathcal{S}$ and $\mathcal{A}$ are the state and action spaces, $P$ is the transition probability function, $R$ is the reward function,

$p_0 \in \Delta(\mathcal{S})$ is the initial state distribution, and $\gamma \in [0, 1]$ is the discount factor. A policy $\pi$ maps states to a distribution over actions: $\pi : \mathcal{S} \mapsto \Delta(\mathcal{A})$, and $\pi(a|s)$ is the probability of choosing action $a$ in state $s$ by policy $\pi$. With a fixed $\pi$, a trajectory $\tau = (s_0, a_0, r_0, s_1, a_1, r_1, \ldots)$ is generated as follows:[1]

$$s_0 \sim p_0(\cdot), \quad a_t \sim \pi(\cdot|s_t), \quad r_t = R(s_t, a_t), \quad s_{t+1} \sim P(\cdot|s_t, a_t), \quad \forall t \geq 0 \,.$$

Given a *target* policy $\pi$, we consider two reward criteria, undiscounted ($\gamma = 1$) and discounted ($\gamma < 1$), where $\mathbb{E}_\pi[\cdot]$ indicates the trajectory $\tau$ is controlled by policy $\pi$:

$$\text{(undiscounted)} \qquad \rho_\pi \quad := \lim_{T \to \infty} \mathbb{E}_\pi \left[ \frac{1}{T} \sum_{t=1}^{T} r_t \right] = \mathbb{E}_{(s,a) \sim d_\pi}[r] \,, \tag{1}$$

$$\text{(discounted)} \qquad \rho_{\pi,\lambda} \quad := (1 - \gamma)\mathbb{E}_\pi \left[ \sum_{t=0}^{\infty} \gamma^t r_t \right] \,. \tag{2}$$

In the above, $d_\pi$ is the stationary distribution over $\mathcal{S} \times \mathcal{A}$, which exists and is unique under certain assumptions (Levin & Peres, 2017).

The $\gamma < 1$ case can be reduced to the undiscounted case of $\gamma = 1$, but not vice versa. Indeed, one can show that the discounted reward in equation 2 can be interpreted as the stationary distribution of an induced Markov process, whose transition function is a mixture of $P$ and the initial-state distribution $p_0$. We refer interested readers to Appendix A for more details. Accordingly, in the following and without the loss of generality, we will merely focus on the more general undiscounted criterion in equation 1, and suppress the unnecessary dependency on $p_0$ and $\gamma$.

In the off-policy estimation problem, we are interested in estimating $\rho_\pi$ for a given target policy $\pi$. However, instead of having access to on-policy trajectories generated by $\pi$, we have a set of $n$ transitions collected by some unknown (i.e., "*black-box*" or behavior-agnostic (Nachum et al., 2019)) behavior mechanism $\pi_{\text{BEH}}$:

$$\mathcal{D} := \{(s_i, a_i, r_i, s_i')\}_{1 \leq i \leq n} \,.$$

Therefore, the goal of off-policy estimation is to estimate $\rho_\pi$ based on $\mathcal{D}$, for a given target policy $\pi$.

The setting we described above is quite general, covering a number of situations. For example, $\pi_{\text{BEH}}$ might be a single policy and $\mathcal{D}$ might consist of one or multiple trajectories collected by $\pi_{\text{BEH}}$. In this special case, $s_i' = s_{i+1}$ for $1 < i < n$; this is the off-policy RL scenario widely studied (e.g., Precup et al., 2001; Sutton et al., 2016; Munos et al., 2016; Liu et al., 2018; Gelada & Bellemare, 2019). Furthermore, if $\pi_{\text{BEH}} = \pi$, we recover the on-policy setting. On the other hand, $\pi_{\text{BEH}}$ and $\mathcal{D}$ can consist of multiple policies and their corresponding trajectories. In this situation, unlike the single policy case $s_i'$ and $s_{i+1}$ might originate from two distinct policies. In general, one can consider $\pi_{\text{BEH}}$ as a distribution over $\mathcal{S} \times \mathcal{A}$ where $(s_i, a_i)$ in $\mathcal{D}$ are sampled from. Having introduced the general setting of the problem, we will describe our estimation approach in the next section.

## 4 BLACK-BOX ESTIMATION

Our estimator is based on the following operator defined on functions over $\mathcal{S} \times \mathcal{A}$. For discrete state-action spaces, given any $d \in \mathbb{R}^{\mathcal{S} \times \mathcal{A}}$,

$$\mathcal{B}_\pi d(s, a) := \pi(a|s) \sum_{\xi \in \mathcal{S}, \alpha \in \mathcal{A}} P(s|\xi, \alpha) d(\xi, \alpha) \,. \tag{3}$$

While we will develop the rest of the paper using the discrete version above for simplicity, the continuous version can be similarly obtained without affecting the estimator and results:

$$\mathcal{B}_\pi d(s, a) = \pi(a|s) \int_{\xi, \alpha} \mathrm{d}P(s|\xi, \alpha) d(\xi, \alpha) \,, \tag{4}$$

where $P$ is now interpreted as the transition kernel.

---

[1] For simplicity in exposition, we assume rewards are deterministic. However, everything in this work generalizes directly to the case of stochastic rewards.

We should note that $\mathcal{B}_\pi$ is indeed different from the Bellman operator (Puterman, 1994); although they have some similarities. In particular, given some state-action pair $(s, a)$, the Bellman operator is defined using next state $s'$ of $(s, a)$, while $\mathcal{B}_\pi$ is defined using *previous* state-actions $(\xi, \alpha)$ that *transition to* $s$. It is in this sense that $\mathcal{B}_\pi$ is backward (in time). Furthermore, as we will show later, $d$ has the interpretation of a distribution over $\mathcal{S} \times \mathcal{A}$. Therefore, $\mathcal{B}_\pi$ describes how visitation *flows* from $(\xi, \alpha)$ to $(s, a)$ and hence, we call it the *backward flow* operator. Note that similar forms of $\mathcal{B}_\pi$ have appeared in the literature, usually used to encode constraints in a dual linear program for an MDP (e.g., Wang et al., 2007; Wang, 2017; Dai et al., 2018). However, the application of $\mathcal{B}_\pi$ for the off-policy estimation problem as considered here appears new to the best of our knowledge.

An important property of $\mathcal{B}_\pi$ is that, under certain assumptions, the stationary distribution $d_\pi$ is the unique fixed point that lies in $\Delta(\mathcal{S} \times \mathcal{A})$ (Levin & Peres, 2017):

$$d_\pi = \mathcal{B}_\pi d_\pi \quad \text{and} \quad d_\pi \in \Delta(\mathcal{S} \times \mathcal{A}) \,. \tag{5}$$

This property is the key element we use to derive our estimator as we describe in the following.

### 4.1 BLACK-BOX ESTIMATOR

In most cases, off-policy estimation involves a weighted average of observed rewards $r_i$ in $\mathcal{D}$. We therefore aim to directly estimate these (non-negative) weights which we denote by $w = \{w_i\} \in \Delta([n])$; that is, $w_i \geq 0$ for $i \in [n]$ and $\sum_{i=1}^n w_i = 1$. Note that the normalization of $w$ may be ensured by techniques such as self-normalized importance sampling (Liu, 2001). Once such a $w$ is obtained, the estimated reward is given by:

$$\hat{\rho}_\pi = \sum_{i=1}^n w_i r_i \,. \tag{6}$$

Effectively, any $w \in \Delta([n])$ defines an empirical distribution which we denote by $d_w$ over $\mathcal{S} \times \mathcal{A}$:

$$d_w(s, a) := \sum_{i=1}^n w_i \mathbf{1}\{s_i = s, a_i = a\} \,. \tag{7}$$

Equation 6 is equivalent to $\hat{\rho}_\pi = \mathbb{E}_{(s,a) \sim d_w}[r]$. Comparing it to equation 1, we naturally want to optimize $w$ so that $d_w$ is close to $d_\pi$. Therefore, inspired by the fixed-point property of $d_\pi$ in equation 5, the problem naturally becomes one of minimizing the discrepancy between $d_w$ and $\mathcal{B}_\pi d_w$. In practice, $w$ is often represented in a parametric way:

$$w_i = \tilde{w}_i / \sum_l \tilde{w}_l \,, \qquad \tilde{w}_i := W(s_i, a_i; \omega) \geq 0 \,, \tag{8}$$

where $W(.)$ is a parametric model, such as neural networks, with parameters $\omega \in \Omega$. We have now reached the following optimization problem:

$$\min_{\omega \in \Omega} \mathbb{D}(d_w \,\|\, \mathcal{B}_\pi d_w) \,, \tag{9}$$

where $\mathbb{D}(\cdot \,\|\, \cdot)$ is some discrepancy function between distributions. In practice, $\mathcal{B}$ is unknown, and must be approximated by samples in the dataset $\mathcal{D}$:

$$\hat{\mathcal{B}}_\pi d_w(s, a) := \pi(a|s) \sum_{i=1}^n w_i \, \mathbf{1}\{s_i' = s\} \,.$$

Clearly, $\hat{\mathcal{B}}_\pi d_w$ is a valid distribution over $\mathcal{S} \times \mathcal{A}$ induced by $w$ and $\mathcal{D}$, and the black-box estimator solves for $w$ by minimizing $\mathbb{D}(d_w \,\|\, \hat{\mathcal{B}}_\pi d_w)$.

### 4.2 BLACK-BOX ESTIMATOR WITH MMD

There are different choices for $\mathbb{D}(\cdot \,\|\, \cdot)$ in equation 9, and multiple approaches to solve it (e.g., Nguyen et al., 2010; Dai et al., 2017). Here, we describe one such algorithm based on Maximum Mean Discrepancy (MMD) (Muandet et al., 2017a). For simplicity, the discussion in this subsection assumes $\mathcal{S} \times \mathcal{A}$ is finite, but the extension to continuous $\mathcal{S} \times \mathcal{A}$ is immediate.

Let $\mathbf{k}(\cdot, \cdot)$ be a positive definite kernel function defined on $(\mathcal{S} \times \mathcal{A})^2$. Given two real-valued functions, $f$ and $g$, defined on $\mathcal{S} \times \mathcal{A}$ we define the following bilinear functional

$$\mathbf{k}[f; \ g] \quad := \sum_{(s,a) \in \mathcal{S} \times \mathcal{A}, (\bar{s}, \bar{a}) \in \mathcal{S} \times \mathcal{A}} f(s,a) \cdot \mathbf{k}\left((s,a), (\bar{s}, \bar{a})\right) \cdot g(\bar{s}, \bar{a}). \tag{10}$$

Clearly, we have $\mathbf{k}[f; \ f] \geq 0$ for any $f$ due to the positive definiteness of $\mathbf{k}$. In addition, $\mathbf{k}$ is called *strictly integrally positive definite* if $\mathbf{k}[f; \ f] = 0$ implies $\|f\|_2 = 0$.

Let $\mathcal{H}$ be the reproducing kernel Hilbert space (RKHS) associated with the kernel function $\mathbf{k}$. This is the unique Hilbert space that includes functions that can be expressed as a sum of countably many terms: $f(\cdot) = \sum_i u^i \ \mathbf{k}((s^i, a^i), \cdot)$, where $\{u^i\} \subset \mathbb{R}$, and $\{(s^i, a^i)\} \subseteq \mathcal{S} \times \mathcal{A}$. The space $\mathcal{H}$ is equipped with an inner product defined as follow: given $f, g \in \mathcal{H}$ such that $f = \sum_i u^i \ \mathbf{k}((s^i, a^i), \cdot)$ and $g = \sum_i v^i \ \mathbf{k}((s^i, a^i), \cdot)$, the inner product is $\langle f, g \rangle_{\mathcal{H}} := \sum_{i,j} u^i v^j \ \mathbf{k}(s^i, a^i), (s^j, a^j))$, which induces the norm defined by $\|f\|_{\mathcal{H}} := \sqrt{\sum_{i,j} u^i u^j \mathbf{k}((s^i, a^i), (s^j, a^j))}$.

Given $\mathcal{H}$, the maximum mean discrepancy between two distributions, $\mu_1$ and $\mu_2$, is defined by

$$\mathbb{D}_{\mathbf{k}}(\mu_1 \| \mu_2) := \sup_{f \in \mathcal{H}} \left\{ \mathbb{E}_{\mu_1}[f] - \mathbb{E}_{\mu_2}[f], \quad \text{s.t.} \quad \|f\|_{\mathcal{H}} \leq 1 \right\}.$$

Here, $f$ may be considered as a discriminator, playing a similar role as the discriminator network in generative adversarial networks (Goodfellow et al., 2014), to measure the difference between $\mu_1$ and $\mu_2$. A useful property of MMD is that it admits a closed-form expression (Gretton et al., 2012):

$$\mathbb{D}_{\mathbf{k}}(\mu_1 \| \mu_2) = \mathbf{k}[\mu_1 - \mu_2; \ \mu_1 - \mu_2]$$
$$= \mathbf{k}[\mu_1; \ \mu_1] - 2\mathbf{k}[\mu_1; \ \mu_2] + \mathbf{k}[\mu_2; \ \mu_2] \ ,$$

where $\mathbf{k}[\cdot; \ \cdot]$ is defined in equation 10, and we used the bilinear property $\mathbf{k}[\cdot; \ \cdot]$. Interested readers are referred to surveys (e.g. Berlinet & Thomas-Agnan, 2011; Muandet et al., 2017b) for more background on RKHS and MMD.

Applying MMD to our objective, we have

$$\mathbb{D}_{\mathbf{k}}(d_w \| \hat{\mathcal{B}}_\pi d_w) \quad = \quad \mathbf{k}[d_w; \ d_w] - 2\mathbf{k}\left[d_w; \ \hat{\mathcal{B}}_\pi d_w\right] + \mathbf{k}\left[\hat{\mathcal{B}}_\pi d_w; \ \hat{\mathcal{B}}_\pi d_w\right] \ .$$

In the above, both $d_w$ and $\hat{\mathcal{B}}_\pi d_w$ are simply probability mass functions on a finite subset of $\mathcal{S} \times \mathcal{A}$, consisting of state-actions encountered in $\mathcal{D}$. It follows immediately from equation 10 that

$$\mathbf{k}[d_w; \ d_w] \quad = \quad \sum_{i,j} w_i w_j \underbrace{k((s_i, a_i), (s_j, a_j))}_{K_{i,j}^{(0)}}$$

$$\mathbf{k}\left[d_w; \ \hat{\mathcal{B}}_\pi d_w\right] \quad = \quad \sum_{i,j} w_i w_j \underbrace{\sum_{a'} \pi(a'|s_j') k((s_i, a_i), (s_j', a'))}_{K_{i,j}^{(1)}}$$

$$\mathbf{k}\left[\hat{\mathcal{B}}_\pi d_w; \ \hat{\mathcal{B}}_\pi d_w\right] \quad = \quad \sum_{i,j} w_i w_j \underbrace{\sum_{a_i', a_j'} \pi(a_i'|s_i')\pi(a_j'|s_j') k((s_i', a_i'), (s_j', a_j'))}_{K_{i,j}^{(2)}} \ .$$

Defining $K_{i,j} := K_{i,j}^{(0)} - 2K_{i,j}^{(1)} + K_{i,j}^{(2)}$, we can express the objective as a function of $\omega$ (since $\{w_i\}$ depends on $\omega$; see equation 8):

$$\ell(\omega) = \sum_{i,j} w_i w_j K_{i,j} \ . \tag{11}$$

**Remark 4.1.** *Mini-batch training is an effective approach to solve large-scale problems. However, the objective $\ell(\omega)$ is not in a form that is ready for mini-batch training, as $w_i$ requires normalization (equation 8) that involves all data in $\mathcal{D}$. Instead, we may equivalently minimize $L(\omega) := \log \ell(\omega)$, which can be turned into a form that allow mini-batch training, using a trick that is also useful in other machine learning contexts (e.g., Jean et al., 2015). See Appendix D for more details.*

Algorithm 1 in Appendix E summarizes our estimator. We next present theoretical analysis of our approach. We show the consistency of our result and provide a sample complexity bound.

### 4.3 THEORETICAL ANALYSIS

**Consistency.** The following theorem shows that the exact minimizer of equation 9 coincides with the fixed point of $\mathcal{B}_\pi$, and the objective function measures the norm of the estimation error in an induced RKHS. To simplify exposition, we assume $x = (s, a)$ and $x' = (s', a')$ a successive action-state pair following $x$: $x' \sim d_\pi(\cdot \mid x)$, where $d_\pi(x' \mid x)$ is the transition probability from $x$ to $x'$, that is, $d_\pi(x' \mid x) = P(s' \mid s, a)\pi(a'|s')$. Similarly, we denote by $(\bar{x}, \bar{x}') = ((\bar{s}, \bar{a}), (\bar{s}', \bar{a}'))$ an independent copy of $(x, x')$.

**Theorem 4.1.** *Suppose* $\mathbf{k}$ *is strictly integrally positive definite, and* $d_\pi$ *is the unique fixed point of* $\mathcal{B}_\pi$ *in equation 5. Then, for any* $d \in \Delta(\mathcal{S} \times \mathcal{A})$,

$$\mathbb{D}_{\mathbf{k}}(d \,||\, \mathcal{B}_\pi d) = 0 \quad \Longleftrightarrow \quad d = d_\pi \,.$$

*Furthermore,* $\mathbb{D}_{\mathbf{k}}(d \,||\, \mathcal{B}_\pi d)$ *equals an MMD between* $d$ *and* $d_\pi$*, with a transformed kernel:*

$$\mathbb{D}_{\mathbf{k}}(d \,||\, \mathcal{B}_\pi d) = \mathbb{D}_{\tilde{\mathbf{k}}}(d \,||\, d_\pi) \,,$$

*where* $\tilde{\mathbf{k}}(x, \bar{x})$ *is a positive definite kernel, defined by*

$$\tilde{\mathbf{k}}(x, \bar{x}) = \mathbb{E}_\pi[\mathbf{k}(x, \bar{x}) - \mathbf{k}(x, \bar{x}') - \mathbf{k}(x', \bar{x}) + \mathbf{k}(x', \bar{x}') \mid (x, \bar{x})],$$

*where the expectation is with respect to* $x' \sim d_\pi(\cdot \mid x)$ *and* $\bar{x}' \sim d_\pi(\cdot \mid \bar{x})$*, with* $x'$ *and* $\bar{x}'$ *drawn independently.*

**Generalization.** We next give a sample complexity analysis. In practice, the estimated weight $\hat{w}$ is based on minimizing the empirical loss $\mathbb{D}_{\mathbf{k}}(d_w \,||\, \hat{\mathcal{B}}_\pi d_w)$, where $\mathcal{B}_\pi$ is replaced by the empirical approximation $\hat{\mathcal{B}}_\pi$. The following theorem compares the empirical weights $\hat{w}$ with the *oracle weight* $w_*$ obtained by minimizing the expected loss $\mathbb{D}_{\mathbf{k}}(d_w \,||\, \mathcal{B}_\pi d_w)$, with the exact transition operator $\mathcal{B}_\pi$.

**Theorem 4.2.** *Assume the weight function is decided by* $w_i = W(s_i, a_i; \omega)/n$*. Denote by* $\mathcal{W} = \{\tilde{W}(\cdot; \omega) \colon \omega \in \Omega\}$ *the model class of* $W(\cdot; \omega)$*. Assume* $\hat{w}$ *is the minimizer of the empirical loss* $\mathbb{D}_{\mathbf{k}}(d_w \,||\, \hat{\mathcal{B}}_\pi d_w)$ *and* $w^*$ *the minimizer of expected loss* $\mathbb{D}_{\mathbf{k}}(d_w \,||\, \mathcal{B}_\pi d_w)$*. Assume* $\{x_i\}_{i=1}^n$ *are i.i.d. samples. Then, with probability* $1 - \delta$ *we have*

$$\mathbb{D}_{\mathbf{k}}(d_{\hat{w}} \,||\, \mathcal{B}_\pi d_{\hat{w}}) - \mathbb{D}_{\mathbf{k}}(d_{w_*} \,||\, \mathcal{B}_\pi d_{w_*}) \leq 16 r_{\max} \mathcal{R}_n(\mathcal{W}) + \frac{16 r_{\max}^2 + r_{\max}^2 \sqrt{8\log(1/\delta)}}{\sqrt{n}},$$

*where* $\mathcal{R}_n(\mathcal{W})$ *denotes the expected Rademacher complexity of* $\mathcal{W}$ *with data size* $n$*, and* $r_{\max} := \max(\|\mathcal{W}\|_\infty, \sup_x \sqrt{\mathbf{k}(x, x)})$*, with* $\|\mathcal{W}\|_\infty := \sup\{\|W\|_\infty : W \in \mathcal{W}\}$*. This suggests a generalization error of* $O(1/\sqrt{n})$ *if* $\mathcal{R}_n(\mathcal{W}) = O(1/\sqrt{n})$*, which is typical for parametric families of functions.*

## 5 EXPERIMENTS

In this section, we present experiments to compare the performance of our proposed method with other baselines on the off-policy evaluation problem. In general and for each experiment, we use a behavior policy $\pi_{\text{BEH}}$ to generate trajectories of length $T_{\text{BEH}}$. We then use these generated samples from a behavior policy to estimate the expected reward of a given target policy $\pi$. To compare our approach with other baselines, we use the root mean squared error (RMSE) with respect to the average long-term reward of the target policy $\pi$. The latter is estimated using a trajectory of length $T_{\text{TAR}} \gg 1$. In particular, we compare our proposed *black-box* approach with the following baselines:

- *naive averaging* baseline in which we simply estimate the expected reward of a target policy by averaging the rewards over the trajectories generated by the behavior policy.
- *model-based* baseline where we use the kernel regression technique with a standard Gaussian RBF kernel. We set the bandwidth of the kernel to the median (or 25[th] or 75[th] percentiles) of the pairwise euclidean distances between the observed data points.
- *inverse propensity score (IPS)* baseline introduced by Liu et al. (2018).

We will first use a simple MDP from Thomas & Brunskill (2016) to highlight the IPS drawback we previously mentioned in Section 1. We then move to classical control benchmarks.

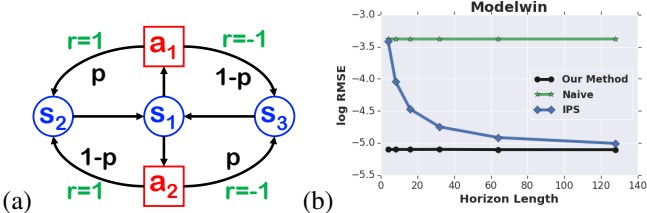

(a)  (b)

Figure 1: (a) ModelWin MDP from Thomas & Brunskill (2016). (b) The RMSE of different methods as we change the length of horizon w.r.t the target policy reward. IPS depends on the horizon length while our method is independent of the horizon length.

## 5.1 TOY EXAMPLE

The *ModelWin* domain first introduced in Thomas & Brunskill (2016) is a fully observable MDP with three states and two actions as denoted in Figure 1(a). The agent always begins in $s_1$ and should choose between two actions $a_1$ and $a_2$. If the agent chooses $a_1$, then with probability of $p$ and $1 - p$ it makes a transition to $s_2$ and $s_3$ and receives a reward of $r = 1$ and $r = -1$, respectively. On the other hand, if the agent chooses $a_2$, then with probability of $p$ and $1 - p$ it makes a transition to $s_3$ with the reward of $r = -1$ and $s_2$ with the reward of $r = 1$, respectively. Once the agent is in either $s_2$ or $s_3$, it goes back to the $s_1$ in the next step without any reward. In our experiments, we set $p = 0.4$.

We define the behavior and target policies as the following. In the target policy, once the agent is in $s_1$, it chooses $a_1$ and $a_2$ with the probability of 0.9 and 0.1, respectively. On the other hand and for the behavior policy, once the agent is in $s_1$, it chooses $a_1$ and $a_2$ with the probability of 0.7 and 0.3, respectively. We calculate the average on-policy reward from samples based on running a trajectory of length $T_{\text{TAR}} = 50,000$ collected by the target policy. We estimate this on-policy reward using trajectories of length $T_{\text{BEH}} \in \{4, 8, 16, 32, 64, 128\}$ collected by the behavior policy. In each case, we set the number of trajectories such that the total number of transitions (i.e., $T_{\text{BEH}}$ times the number of trajectories) is kept constant. For example, for $T_{\text{BEH}} = 4$ and $T_{\text{BEH}} = 8$ we use 50,000 and 25,000 trajectories, respectively. Since the problem has finitely many state-actions, we use the tabular method and hence, equation 11 turns into a quadratic programming. We then report the result of each setting based on 10 Monte-Carlo samples.

As we can see in Figure 1(b), the naive averaging method performs poorly consistently and independent of the length of trajectories collected by the behavior policies. On the other hand, IPS performs poorly when the collected trajectories have short-horizon and gets better as the horizon length of trajectories get larger. This is expected for IPS — as mentioned in Section 1, it requires data be drawn from the stationary distribution. In contrast, as shown in Figure 1(b), our black-box approach performance is independent of the horizon length, and substantially better.

## 5.2 CLASSIC CONTROL

We now focus on four classic control problems. We begin by briefly describing each problem and then compare the performance of our method with other approaches on these problems. Note that for these problems are episodic, we convert them into infinite-horizon problems by resetting the state to a random start state once the episode terminates.

**Pendulum.** In this environment, the goal is to control a pendulum in a vertical position. State variables are the pole angle $\theta$ and velocity $\dot{\theta}$. The action $a$ is the torque in the set $\{-2, -1, 0, 1, 2\}$ applied to the base. We set the reward function to $-(\theta^2 + 0.1\dot{\theta}^2 + 0.001a^2)$.

**Mountain Car.** For this problem, the goal is to drive up the car to top of a hill. Mountain Car has a state space of $\mathbb{R}^2$ (the position and speed of the car) and three possible actions (negative, positive, or zero acceleration). We set the reward to +100 when the car reaches the goal and -1 otherwise.

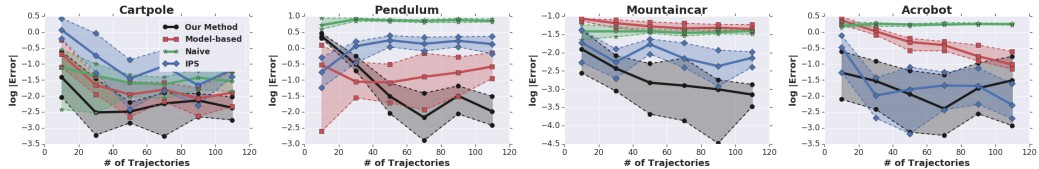

Figure 2: The RMSE of different methods w.r.t the target policy reward as we change the number of trajectories. Our black-box approach outperforms other methods on three problems.

Figure 3: The median and error bars at 25$^{\text{th}}$ and 75$^{\text{th}}$ percentiles of different methods w.r.t the target policy reward as we change the number of trajectories. The trend of results is similar to Figure 2.

**Cartpole.** The goal here is to prevent an attached pole to a cart from falling by changing the cart's velocity. Cartpole has a state space of $\mathbb{R}^4$ (cart position, velocity, pole angle and velocity) and two possible actions (moving left or right). Reward is -100 when the pole falls and +1 otherwise.

**Acrobot.** In this problem, our goal is to swing a 2-link pendulum above the base. Acrobot has a state space of $\mathbb{R}^6$ ($\sin(.)$ and $\cos(.)$ of both angles and angular velocities) and three possible actions (applying +1, 0 or -1 torque on the joint). Reward is +100 for reaching the goal and -1 otherwise.

For each environment, we train a near-optimal policy $\pi_+$ using the *Neural Fitted Q Iteration* algorithm (Riedmiller, 2005). We then set the behavior and target policies as $\pi_{\text{BEH}} = \alpha_1\pi_+ + (1 - \alpha_1)\pi_-$ and $\pi = \alpha_2\pi_+ + (1 - \alpha_2)\pi_-$, where $\pi_-$ denotes a random policy, and $0 \leq \alpha_1, \alpha_2 \leq 1$ are two constant values making the behavior policy distinct from the target policy. In our experiments, we set $\alpha_1 = 0.7$ and $\alpha_2 = 0.9$. In order to calculate the on-policy reward, we use a single trajectory collected by $\pi$ with $T_{\text{TAR}} = 50,000$. For off-policy data, we use multiple trajectories collected by $\pi_{\text{BEH}}$ with $T_{\text{BEH}} = 200$. In all the cases, we use a 3-layer (having 30, 20, and 10 hidden neurons) feed-forward neural network with the sigmoid activation function as our parametric model in equation 8. For each setting, we report results based on 20 Monte-Carlo samples.

Figure 2 shows the $\log$ of RMSE w.r.t. the target policy reward as we change the number of trajectories collected by the behavior policy. We should note that all methods except the naive averaging method have hyperparameters to be tuned. For each method, the optimal set of parameters might depend on the number of trajectories (i.e., size of the training data). However, in order to avoid excessive tuning and to show how much each method is robust to a change in the number of trajectories, we only tune different methods based on 50 trajectories and use the same set of parameters for other settings. As we can see, the naive averaging performance is almost independent of the number of trajectories. Our method outperforms other approaches on three environments and it is only the Acrobot in which IPS performs comparably to our black-box approach. In order to have a robust evaluation against outliers, we have plotted the median and error bars at 25$^{\text{th}}$ and 75$^{\text{th}}$ percentiles in Figure 3. If we compare the Figures 2 and 3, we notice that the trend of results is almost the same in both. In Appendix G, we have studied how changing $\alpha_1$ of the behavior policy affects the final RMSE.

## 6 CONCLUSIONS

In this paper, we presented a novel approach for solving the off-policy estimation problem in the long-horizon setting. Our method formulates the problem as solving for the fixed point of a "backward flow" operator. We showed that unlike previous works, our approach does not require the knowledge of the behavior policy or stationary off-policy data. We presented experimental results to show the effectiveness of our approach compared to previous baselines. In the future, we plan to use structural domain knowledge to improve the estimator and consider a random time horizon in episodic RL.

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

## A    REDUCTION FROM DISCOUNTED TO UNDISCOUNTED REWARD

The same reduction is used in Liu et al. (2018). For completeness, we give the derivation details here, for the case of finite state/actions. The derivation can be extended to general state-action spaces, with proper adjustments in notation.

Let $\tau = (s_0, a_0, r_0, s_1, a_1, \ldots)$ be a trajectory generated by $\pi$, and $d_t \in \Delta(\mathcal{S} \times \mathcal{A})$ be the distribution of $(s_t, a_t)$. Clearly,

$$
\begin{aligned}
d_0(s, a) &= p_0(s)\pi(a|s) \\
d_{t+1}(s, a) &= \sum_{\xi, \alpha} d_t(\xi, \alpha) P(s|\xi, \alpha)\pi(a|s), \qquad \forall t > 0 \,.
\end{aligned}
$$

Using a matrix form, the recursion above can be written equivalently as $d_{t+1} = P_\pi^{\mathrm{T}} d_t$, where $P_\pi$ is given by

$$
P_\pi(s, a|\xi, \alpha) = P(s|\xi, \alpha)\pi(a|s) \,.
$$

The discounted reward of policy $\pi$ is

$$
\rho_{\pi,\gamma} = (1 - \gamma)\mathbb{E}_\pi \left[ \sum_{t=0}^{\infty} \gamma^t r_t \right] = \mathbb{E}_{(s,a)\sim d_{\pi,\gamma}}[R(s, a)] \,,
$$

with

$$
d_{\pi,\gamma} := (1 - \gamma)\left(d_0 + \gamma d_1 + \gamma^2 d_2 + \cdots\right) \,.
$$

Multiplying both sides of above by $\gamma P_\pi^{\mathrm{T}}$, we have

$$
\begin{aligned}
\gamma P_\pi^{\mathrm{T}} d_{\pi,\gamma} &= (1 - \gamma)\left(\gamma P_\pi^{\mathrm{T}} d_0 + \gamma^2 P_\pi^{\mathrm{T}} d_1 + \gamma^3 P_\pi^{\mathrm{T}} d_2 + \cdots\right) \\
&= (1 - \gamma)\left(\gamma d_1 + \gamma^2 d_2 + \gamma^3 d_3 + \cdots\right) \\
&= d_{\pi,\gamma} - (1 - \gamma)d_0 \,.
\end{aligned}
$$

Therefore,

$$
\begin{aligned}
d_{\pi,\gamma} &= \gamma P_\pi^{\mathrm{T}} d_{\pi,\gamma} + (1 - \gamma)d_0 \\
&= (\gamma P_\pi + (1 - \gamma)d_0 \mathbf{1}^{\mathrm{T}})^{\mathrm{T}} d_{\pi,\gamma} \,.
\end{aligned}
$$

Accordingly, $d_{\pi,\gamma}$ is the fixed point of an induced transition matrix given by $P_{\pi,\lambda} := \gamma P_\pi + (1 - \gamma)d_0 \mathbf{1}^{\mathrm{T}}$. This completes the reduction from the discounted to the undiscounted case.

## B    PROOF OF THEOREM 4.1

Note that

$$
\mathbb{D}_{\mathbf{k}}(d \,\|\, \mathcal{B}_\pi d) = \mathbf{k}\left[d - \mathcal{B}_\pi d;\ d - \mathcal{B}_\pi d\right] \,.
$$

Following the definition of the strictly integrally positive definite kernels, we have that $\mathbb{D}_{\mathbf{k}}(d \,\|\, \mathcal{B}_\pi d) = 0$ implies $d - \mathcal{B}_\pi d = 0$, which in turn implies $d = d_\pi$ by the uniqueness assumption of $d_\pi$. We have thus proved the first claim.

For the second claim, define $\delta_w = d - d_\pi$. Since $d_\pi - \mathcal{B}_\pi d_\pi = 0$, we have

$$
\begin{aligned}
\mathbb{D}_{\mathbf{k}}(d \,\|\, \mathcal{B}_\pi d) &= \mathbf{k}\left[(d - \mathcal{B}_\pi d);\ (d - \mathcal{B}_\pi d)\right] \\
&= \mathbf{k}\left[(d - \mathcal{B}_\pi d - (d_\pi - \mathcal{B}_\pi d_\pi));\ (d - \mathcal{B}_\pi d - (d_\pi - \mathcal{B}_\pi d_\pi))\right] \\
&= \mathbf{k}\left[(\delta_w - \mathcal{B}_\pi \delta_w);\ (\delta_w - \mathcal{B}_\pi \delta_w)\right] \,.
\end{aligned}
$$

Recalling that $\mathcal{B}_\pi d(x) = \sum_{x_0} P_\pi(x|x_0)d(x_0)$, we have

$$
\begin{aligned}
\mathbb{D}_{\mathbf{k}}(d \,\|\, \mathcal{B}_\pi d) &= \mathbf{k}\left[(\delta_w - \mathcal{B}_\pi \delta_w);\ (\delta_w - \mathcal{B}_\pi \delta_w)\right] \\
&= \sum_{x, \bar{x}} \mathbf{k}(x, \bar{x})(\delta_w(x) - \mathcal{B}_\pi \delta_w(x))(\delta_w(\bar{x}) - \mathcal{B}_\pi \delta_w(\bar{x})) \\
&= \sum_{x, \bar{x}} \mathbf{k}(x, \bar{x})\left(\delta_w(x) - \sum_{x_0} P_\pi(x|x_0)\delta_w(x_0)\right)\left(\delta_w(\bar{x}) - \sum_{\bar{x}_0} P_\pi(\bar{x}|\bar{x}_0)\delta_w(\bar{x}_0)\right) \,.
\end{aligned}
$$

Define the adjoint operator of $\mathcal{B}_\pi$,

$$\mathcal{P}_\pi f(x) := \sum_{x'} P_\pi(x'|x) f(x').$$

Denote by $\mathcal{P}_\pi^x$ the operator applied on $\mathbf{k}(x, \bar{x})$ in terms of variable $x$, that is, $\mathcal{P}_\pi^x \mathbf{k}(x, \bar{x}) := \sum_{x'} P_\pi(x'|x) \mathbf{k}(x', \bar{x})$. This gives

$$\mathbb{D}_\mathbf{k}(d \parallel \mathcal{B}_\pi d) = \sum_{x,x'} \mathbf{k}(x, x') \left( \delta_w(x) - \sum_{x_0} P_\pi(x|x_0) \delta_w(x_0) \right) \left( \delta_w(\bar{x}) - \sum_{\bar{x}_0} P_\pi(\bar{x}|\bar{x}_0) \delta_w(\bar{x}_0) \right)$$

$$= \sum_{x,\bar{x}} \delta_w(x) \left( \mathbf{k}(x, \bar{x}) - \mathcal{P}_\pi^x \mathbf{k}(x, \bar{x}) - \mathcal{P}_\pi^{\bar{x}} \mathbf{k}(x, \bar{x}) + \mathcal{P}_\pi^x \mathcal{P}_\pi^{\bar{x}} \mathbf{k}(x, \bar{x}) \right) \delta(\bar{x})$$

$$= \sum_{x,\bar{x}} \delta_w(x) \tilde{\mathbf{k}}_\pi(x, \bar{x}) \delta_w(\bar{x}) \,,$$

completing the proof.

## C  PROOF OF THEOREM 4.2

First, note that the error can be decomposed as follows:

$$\mathbb{D}_\mathbf{k}(d_{\hat{w}} \parallel \mathcal{B}_\pi d_{\hat{w}}) \leq \mathbb{D}_\mathbf{k}(d_{\hat{w}} \parallel \hat{\mathcal{B}}_\pi d_{\hat{w}}) + \mathbb{D}_\mathbf{k}(\hat{\mathcal{B}}_\pi d_{\hat{w}} \parallel \mathcal{B}_\pi d_{\hat{w}})$$

$$\leq \mathbb{D}_\mathbf{k}(d_{w_*} \parallel \hat{\mathcal{B}}_\pi d_{w_*}) + \mathbb{D}_\mathbf{k}(\hat{\mathcal{B}}_\pi d_{\hat{w}} \parallel \mathcal{B}_\pi d_{\hat{w}})$$

$$\leq \mathbb{D}_\mathbf{k}(d_{w_*} \parallel \mathcal{B}_\pi d_{w_*}) + \mathbb{D}_\mathbf{k}(\hat{\mathcal{B}}_\pi d_{w_*} \parallel \mathcal{B}_\pi d_{w_*}) + \mathbb{D}_\mathbf{k}(\hat{\mathcal{B}}_\pi d_{\hat{w}} \parallel \mathcal{B}_\pi d_{\hat{w}})$$

$$\leq \mathbb{D}_\mathbf{k}(d_{w_*} \parallel \mathcal{B}_\pi d_{w_*}) + 2 \sup_{w \in \mathcal{W}} \mathbb{D}_\mathbf{k}(\hat{\mathcal{B}}_\pi d_w \parallel \mathcal{B}_\pi d_w)$$

$$= \mathbb{D}_\mathbf{k}(d_{w_*} \parallel \mathcal{B}_\pi d_{w_*}) + 2Z,$$

where we define

$$Z := \mathbb{D}_\mathbf{k}(\hat{\mathcal{B}}_\pi d_w \parallel \mathcal{B}_\pi d_w).$$

Therefore, we just need to bound $Z$.

Denote by $\mathcal{B}_\mathbf{k} := \{f \colon f \in \mathcal{H}, \; \|f\|_\mathcal{H} \leq 1\}$ the unit ball of RKHS. Define $\|\mathcal{B}_\mathbf{k}\| := \sup_{f \in \mathcal{B}_\mathbf{k}}$ and $\mathcal{R}_n(\mathcal{B}_\mathbf{k})$ the expected Rademacher complexity of $\mathcal{B}_\mathbf{k}$ of data size $n$. From classical RKHS theory (see Lemma C.2 below), we know that $\|\mathcal{B}_\mathbf{k}\|_\infty \leq r_\mathbf{k}$ and $\mathcal{R}_n(\mathcal{B}_\mathbf{k}) \leq r_\mathbf{k}/\sqrt{n}$, where $r_\mathbf{k} := \sqrt{\sup_{x \in \mathcal{X}} \mathbf{k}(x, x')}$.

It remains to calculate $Z := \sup_{w \in \mathcal{W}} \mathbb{D}_\mathbf{k}(\hat{\mathcal{B}}_\pi d_w \parallel \mathcal{B}_\pi d_w))$. Recall from the definition of $\mathbb{D}_\mathbf{k}$ that

$$\mathbb{D}_\mathbf{k}(\hat{\mathcal{B}}_\pi d_w \parallel \mathcal{B}_\pi d_w)) = \sup_{f \in \mathcal{B}_\mathbf{k}} \mathbb{E}_{x \sim \hat{\mathcal{B}}_\pi d_w}[f(x)] - \mathbb{E}_{x \sim \mathcal{B}_\pi d_w}[f(x)] \,.$$

Recall that $\{(x_i, x_i')\}_{i=1}^n$ is a set of transitions with $x_i' \sim d_\pi(\cdot \mid x_i)$, where $d_\pi$ denotes the transition probability under policy $\pi$. Following the definitions of $\hat{\mathcal{B}}_\pi d_w$ and $\mathcal{B}_\pi d_w$, we have

$$\hat{\mathcal{B}}_\pi d_w(x) = \sum_{i=1}^n w(x_i) \mathbf{1}\{x = x_i'\} \,,$$

$$\mathcal{B}_\pi d_w(x) = \sum_{i=1}^n w(x_i) d_\pi(x \mid x_i) \,.$$

Therefore,

$$Z = \sup_{w \in \mathcal{W}} \mathbb{D}_\mathbf{k}(\hat{\mathcal{B}}_\pi d_w \parallel \mathcal{B}_\pi d_w))$$

$$= \sup_{w \in \mathcal{W}} \sup_{f \in \mathcal{B}_\mathbf{k}} \mathbb{E}_{x \sim \hat{\mathcal{B}}_\pi d_w}[f(x)] - \mathbb{E}_{x \sim \mathcal{B}_\pi d_w}[f(x)]$$

$$= \sup_{w \in \mathcal{W}, f \in \mathcal{B}_\mathbf{k}} \frac{1}{n} \sum_{i=1}^n w(x_i) \left( f(x_i') - \mathbb{E}_{\bar{x}_i' \sim d_\pi}[f(\bar{x}_i')|x_i] \right) \,,$$

where $\bar{x}_i' \sim d_\pi(\cdot \mid x_i)$ is an independent copy of $x_i'$ that follows the same distribution. Note that $\bar{x}_i'$ is introduced only for the sample complexity analysis. Note that $Z$ is a random variable, and $\mathbb{E}[Z]$ denotes its expectation w.r.t. random data $\{x_i, x_i'\}_{i=1}^n$. We assume different $(x_i, x_i')$ are independent with each other. First, by McDiarmid inequality, we have

$$P(Z \geq \mathbb{E}[Z] + \epsilon) \leq \exp\left(-\frac{n\epsilon^2}{2\|\mathcal{W}\|_\infty^2 \|\mathcal{B}_\mathbf{k}\|_\infty^2}\right).$$

This is because when changing each data point $(x_i, x_i')$, the maximum change on $Z$ is at most $2\|\mathcal{W}\|_\infty \|\mathcal{B}_\mathbf{k}\|_\infty /n$. Therefore, we have $Z \leq \mathbb{E}[Z] + \sqrt{2\log(1/\delta)\|\mathcal{W}\|_\infty^2 \|\mathcal{B}_\mathbf{k}\|_\infty^2 /n}$ with probability at least $1 - \delta$.

Accordingly, we now just need to bound $\mathbb{E}[Z]$:

$$\begin{aligned}
\mathbb{E}[Z] &= \mathbb{E}_X\left[\sup_{w\in\mathcal{W}, f\in\mathcal{B}_\mathbf{k}} \frac{1}{n}\sum_{i=1}^n w(x_i)\left(f(x_i') - \mathbb{E}_{\bar{X}}[f(\bar{x}_i')|x_i]\right)\right] \\
&\leq \mathbb{E}_{X,\bar{X}}\left[\sup_{w\in\mathcal{W}, f\in\mathcal{B}_\mathbf{k}} \frac{1}{n}\sum_{i=1}^n w(x_i)(f(x_i') - f(\bar{x}_i'))\right] \\
&= \mathbb{E}_{X,\bar{X},\sigma}\left[\sup_{w\in\mathcal{W}, f\in\mathcal{B}_\mathbf{k}} \frac{1}{n}\sum_{i=1}^n \sigma_i w(x_i)(f(x_i') - f(\bar{x}_i'))\right] \\
&\qquad \text{(because } \{\sigma_i\} \text{ are i.i.d. Rademacher random variables)} \\
&\leq 2\mathbb{E}\left[\sup_{w\in\mathcal{W}, f\in\mathcal{B}_\mathbf{k}} \frac{1}{n}\sum_{i=1}^n \sigma_i w(x_i) f(x_i')\right] \\
&= 2\mathcal{R}_n(\mathcal{W} \otimes \mathcal{B}_\mathbf{k}),
\end{aligned}$$

where

$$\mathcal{W} \otimes \mathcal{B}_\mathbf{k} = \{f(x)g(x') : f \in \mathcal{W}, \ g \in \mathcal{B}_\mathbf{k}\}.$$

By Lemma C.1 below, we have

$$\mathbb{E}[Z] \leq 2\mathcal{R}_n(\mathcal{W} \otimes \mathcal{B}_\mathbf{k}) \leq 4\left(\|\mathcal{W}\|_\infty + \|\mathcal{B}_\mathbf{k}\|_\infty\right)\left(\mathcal{R}_n(\mathcal{W}) + \mathcal{R}_n(\mathcal{B}_\mathbf{k})\right).$$

Combining the bounds, we have with probability $1 - \delta$,

$$2Z \leq 4\mathcal{R}_n(\mathcal{W} \otimes \mathcal{B}_\mathbf{k}) + \sqrt{\frac{8\log(1/\delta)\|\mathcal{W}\|_\infty^2 \|\mathcal{B}_\mathbf{k}\|_\infty^2}{n}}$$

$$\leq 8\left(\|\mathcal{W}\|_\infty + \|\mathcal{B}_\mathbf{k}\|_\infty\right)\left(\mathcal{R}_n(\mathcal{W}) + \mathcal{R}_n(\mathcal{B}_\mathbf{k})\right) + \sqrt{\frac{8\log(1/\delta)\|\mathcal{W}\|_\infty^2 \|\mathcal{B}_\mathbf{k}\|_\infty^2}{n}}.$$

Plugging Lemma C.2, we have

$$2Z \leq 8\left(\|\mathcal{W}\|_\infty + r_\mathbf{k}\right)\mathcal{R}_n(\mathcal{W}) + \frac{8r_\mathbf{k}\left(\|\mathcal{W}\|_\infty + r_\mathbf{k} + \|\mathcal{W}\|_\infty\sqrt{\log(1/\delta)/8}\right)}{\sqrt{n}}.$$

Assume $r_{\max} = \max(\|\mathcal{W}\|_\infty, r_\mathbf{k})$. We have

$$2Z \leq 16\, r_{\max}\mathcal{R}_n(\mathcal{W}) + \frac{16r_{\max}^2 + r_{\max}^2\sqrt{8\log(1/\delta)}}{\sqrt{n}}.$$

**Lemma C.1.** *Denote by $\|\mathcal{W}\|_\infty = \sup\{\|f\|_\infty : f \in \mathcal{W}\}$ the super norm of a function set $\mathcal{W}$. Then,*

$$\mathcal{R}_n(\mathcal{W} \otimes \mathcal{B}_\mathbf{k}) \leq 2\left(\|\mathcal{W}\|_\infty + \|\mathcal{B}_\mathbf{k}\|_\infty\right)\left(\mathcal{R}_n(\mathcal{W}) + \mathcal{R}_n(\mathcal{B}_\mathbf{k})\right).$$

*Proof.* Note that

$$f(x)g(x') = \frac{1}{4}(f(x) + g(x'))^2 - \frac{1}{4}(f(x) - g(x'))^2.$$

Note that $x^2$ is $2(\|\mathcal{W}\|_\infty + \|\mathcal{B}_\mathbf{k}\|_\infty)$-Lipschitz on interval $[-\|\mathcal{W}\|_\infty - \|\mathcal{B}_\mathbf{k}\|_\infty, \|\mathcal{W}\|_\infty + \|\mathcal{B}_\mathbf{k}\|_\infty]$. Applying Lemma C.1 of Liu & Wang (2018), we have

$$\mathcal{R}_n(\mathcal{W} \otimes \mathcal{B}_\mathbf{k}) \le 2(\|\mathcal{W}\|_\infty + \|\mathcal{B}_\mathbf{k}\|_\infty)(R_n(\mathcal{W} \oplus \mathcal{B}_\mathbf{k}),$$

where $\mathcal{W} \oplus \mathcal{B}_\mathbf{k} = \{f(x) + g(x') \colon f \in \mathcal{W}, \ g \in \mathcal{B}_\mathbf{k}\}$, and

$$\mathcal{R}_n(\mathcal{W} \oplus \mathcal{B}_\mathbf{k}) = \mathbb{E}_{\boldsymbol{v}z}[\sup_{f \in \mathcal{W}, g \in \mathcal{B}_\mathbf{k}} \sum_i z_i(f(x_i) + g(x_i'))]$$

$$\le \mathbb{E}_{\boldsymbol{v}z}[\sup_{f \in \mathcal{W}} \sum_i z_i f(x_i)] + \mathbb{E}_{\boldsymbol{v}z}[\sup_{g \in \mathcal{B}_\mathbf{k}} \sum_i z_i g(x_i')]$$

$$= \mathcal{R}_n(\mathcal{W}) + \mathcal{R}_n(\mathcal{B}_\mathbf{k}).$$

$\square$

**Remark** The same result holds when $w$ is defined as a function of the whole transition pair $(x, x')$, that is, $w_i = w(x_i, x_i')$.

**Lemma C.2.** *Let $\mathcal{H}$ be the RKHS with a positive definite kernel $\mathbf{k}(x, x')$ on domain $\mathcal{X}$. Let $\mathcal{B}_\mathbf{k} = \{f \in \mathcal{H}_\mathbf{k} \colon \|f\|_{\mathcal{H}_\mathbf{k}} \le 1\}$ be the unit ball of $\mathcal{H}_\mathbf{k}$, and $r_\mathbf{k} = \sqrt{\sup_{x \in \mathcal{X}} \mathbf{k}(x, x')}$. Then,*

$$\|\mathcal{B}_\mathbf{k}\|_\infty \le r_\mathbf{k}, \qquad \mathcal{R}_n(\mathcal{B}_\mathbf{k}) \le \frac{r_\mathbf{k}}{\sqrt{n}}.$$

*Proof.* These are standard results in RKHS theory, and we give a proof for completeness. For $\|\mathcal{B}_\mathbf{k}\|_\infty$, we just note that for any $f \in \mathcal{B}_\mathbf{k}$ and $x \in \mathcal{X}$,

$$f(x) = \langle f, \ \mathbf{k}(x, \cdot) \rangle_{\mathcal{H}_\mathbf{k}} \le \|f\|_{\mathcal{H}_\mathbf{k}} \|\mathbf{k}(x, \cdot)\|_{\mathcal{H}_\mathbf{k}} \le \|\mathbf{k}(x, \cdot)\|_{\mathcal{H}_\mathbf{k}} = \sqrt{\mathbf{k}(x, x)} \le r_\mathbf{k}.$$

The Rademacher complexity can be bounded as follows:

$$\mathcal{R}_n(\mathcal{B}_\mathbf{k}) = \mathbb{E}_{X, \sigma} \left[ \sup_{f \in \mathcal{B}_\mathbf{k}} \frac{1}{n} \sum_i \sigma_i f(x_i) \right]$$

$$\le \mathbb{E}_{X, \sigma} \left[ \sup_{f \in \mathcal{B}_\mathbf{k}} \left\langle f, \frac{1}{n} \sum_{i=1}^n \sigma_i \mathbf{k}(x_i, \cdot) \right\rangle_{\mathcal{H}_\mathbf{k}} \right]$$

$$= \mathbb{E}_{X, \sigma} \left[ \left\| \frac{1}{n} \sum_{i=1}^n \sigma_i \mathbf{k}(x_i, \cdot) \right\|_{\mathcal{H}_\mathbf{k}} \right]$$

$$\le \mathbb{E}_{X, \sigma} \left[ \left\| \frac{1}{n} \sum_{i=1}^n \sigma_i \mathbf{k}(x_i, \cdot) \right\|_{\mathcal{H}_\mathbf{k}}^2 \right]^{1/2}$$

$$= \mathbb{E}_{X, \sigma} \left[ \frac{1}{n^2} \sum_{i,j=1}^n \sigma_i \sigma_j \mathbf{k}(x_i, x_j) \right]^{1/2}$$

$$= \mathbb{E}_X \left[ \frac{1}{n^2} \sum_{i=1}^n \mathbf{k}(x_i, x_i) \right]^{1/2}$$

$$\le \frac{r_\mathbf{k}}{\sqrt{n}}.$$

$\square$

## D    Mini-batch training

The objective $\ell(\omega)$ is not in a form that is ready for mini-batch training. It is possible to yield better scalability with a trick that has been found useful in other machine learning contexts (e.g., Jean et al., 2015). We start with a transformed objective:

$$
\begin{aligned}
L(\omega) & := \log \ell(\omega) \\
& = \log \sum_{i,j} \tilde{w}_i \tilde{w}_j K_{ij} - 2 \log \sum_l \tilde{w}_l \, .
\end{aligned}
$$

Then,

$$
\begin{aligned}
\nabla L & = \frac{\sum_{i,j} \nabla(\tilde{w}_i \tilde{w}_j) K_{ij}}{\sum_{uv} \tilde{w}_u \tilde{w}_v K_{uv}} - \frac{2 \sum_i \nabla \tilde{w}_i}{\sum_l \tilde{w}_l} \\
& = \frac{\sum_{i,j} \tilde{w}_i \tilde{w}_j K_{ij} \nabla \log(\tilde{w}_i \tilde{w}_j)}{\sum_{uv} \tilde{w}_u \tilde{w}_v K_{uv}} - \frac{2 \sum_i \tilde{w}_i \nabla \log \tilde{w}_i}{\sum_l \tilde{w}_l} \\
& = \hat{\mathbb{E}}_{ij}[\nabla \log(\tilde{w}_i \tilde{w}_j)] - \hat{\mathbb{E}}_i[\nabla \log \tilde{w}_i] \, ,
\end{aligned}
$$

where $\hat{\mathbb{E}}_{ij}[\cdot]$ and $\hat{\mathbb{E}}_i[\cdot]$ correspond to two properly defined discrete distributions defined on $\mathcal{D}^2$ and $\mathcal{D}$, respectively. Clearly, $\nabla L$ can now be approximated by mini-batches by drawing random samples from $\mathcal{D}^2$ or $\mathcal{D}$ to approximate $\hat{E}_{ij}$ and $\hat{E}_i$.

## E    Pseudo-Code of Algorithm

This section includes the pseudo-code of our algorithm that we described in Section 4.

---
**Algorithm 1** Black-box Off-policy Estimator based on MMD
---
**Inputs:** Dataset $\mathcal{D} = \{(s_i, a_i, r_i, s'_i)\}_{1 \leq i \leq n}$ of behavior policy $\pi_{\text{BEH}}$, kernel function $\mathbf{k}$, target policy $\pi$, parametric model $W(.)$.
**Output:** Target policy value estimate $\hat{\rho}_\pi$.
  1: Estimate important weights $\{w_i\}_{1 \leq i \leq n}$ by solving the optimization problem in equation 11.
  2: **return** $\hat{\rho}_\pi = \sum_{i=1}^n w_i r_i$ (see equation 6).
---

## F    Bias-Variance Comparison

In this section, we compare the performance of our approach from the bias and variance perspective. In particular, we focus on the ModelWin MDP (Figure 1). In order to see the impact of training data size on the performance of different estimators, we consider different number of trajectories of length 4. For each setting, we have 200 independent runs and calculate the bias and variance of each method based on them. We compare our approach with Liu et al. (2018) and Nachum et al. (2019) that are two state-of-the-art methods in the off-policy estimation problem. Figure 4 shows the results of this comparison.

As we can see, our method has a smaller bias but larger variance compared to other approaches on this problem. As mentioned before, DualDICE (Nachum et al., 2019) does not cover the undiscounted reward criterion. Therefore, instead of $\gamma = 1$, we have used $\gamma = 0.9999$ in the code shared by the authors. This induces some bias but reduces the variance of this method. To confirm this observation, we further decreased the $\gamma$ to 0.9 in DualDICE and observed that reducing $\gamma$ in this algorithm increases the bias but reduces the variance at the same time.

The comparison to IPS of Liu et al. (2018) highlights the significance of an assumption needed by IPS: the data must be drawn (approximately) from the stationary distribution of the behavior policy. As the trajectory length is 4 in the experiment, this assumption is apparently violated, thus the high bias of IPS. In contrast, our method does not rely on such an assumption, so is more robust.

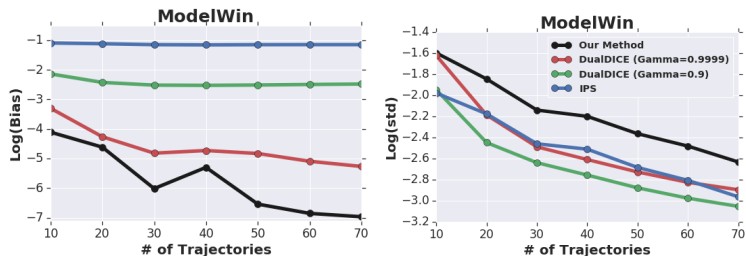

Figure 4: Bias and standard deviation of different methods for the ModelWin MDP (Fig. 1). Our method has a smaller bias but larger variance compared to other algorithms.

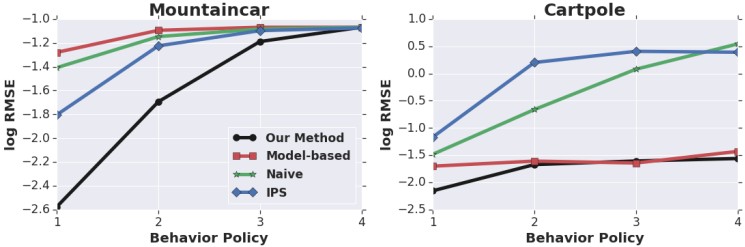

Figure 5: The RMSE of different methods w.r.t the target policy reward as we change the behavior policy. Our method outperform other approaches on different behavior policies.

## G  SENSITIVITY

Finally, in Figure 5 we measure how robust our approach is to changing the behavior policy compared to other methods. In particular, we vary $\alpha_1$ that corresponds to the behavior policy to measure how the RMSE is affected. While $\alpha_2$ is fixed to 0.9, in each experiment we choose $\alpha_1$ from $\{0.7, 0.5, 0.3, 0.1\}$. For each experiment, we use data from 50 trajectories (with $T_{\text{BEH}} = 200$) collected by the behavior policy and report results based on 20 Monte-Carlo samples. According to Figure 5, as $\alpha_1$ diverges more from $\alpha_2$, the performance of all the methods degrade while our method is the least affected. It is worth mentioning that for the Mountain Car problem and $\alpha_1 = 0.1$, the behavior policy is close to a random policy and hence the car has not been able to drive up to top of the hill. This means that all the methods have constantly received a reward of $-1$ during all the steps and hence the estimated on-policy reward has been -1 for all the methods as well. Therefore, the RMSE of all four methods are equal in this case.

