# OpenReview forum: "Black-box Off-policy Estimation for Infinite-Horizon Reinforcement Learning"
_ICLR.cc/2020/Conference — Accept (Poster)_

### Official Review · AnonReviewer2 · 2019-10-21
**Official Blind Review #2**

**Rating:** 6

**Review:**

Summary

This paper proposes using a black-box estimation method from Liu and Lee 2017 to estimate the propensity score for off-policy reinforcement learning. They suggest that, even when the full proposal distribution is available (as is the case in much of the off-policy RL literature), that estimating the propensity score can allow for a much lower variance off-policy correction term. This reduced variance allows the proposed method to "break the curse of horizon" as termed by Liu 2018.

Review

The proposed fixed point formulation for learning a parameterized off-policy state distribution allows for lower variance off-policy corrections decreasing the impact of the curse of horizon. This approach appears both theoretically sound and empirically well-supported.

I am concerned with the consistency argument made in section 4.3. It appears from equation (10) and the final statement of theorem 4.1 that it is necessary to have two independent samples of x' given a single state x. In the general case, without access to the environment model, it is not possible to obtain two samples of x'. If the environment is continuing, then the probability of returning to state x to obtain a second sample is 0. Am I misunderstanding the requirements specified by the objective function in equation (10)? An additional concern with the consistency argument is that it appears to assume that the approximator for d_w can achieve 0 error according to the MMD. If d_pi is not representable by the approximator, which could reasonably be the case in more challenging domains, it is unclear from this analysis or empirical results what the behavior of the system will be. It is difficult to assess how large of an assumption this is for consistency claim; for difficult problems will d_pi never be representable, or is this a fairly low concern?

The empirical results look to have high enough variance in the final outcomes that it is difficult to consistently assess the performance of each algorithm (looking specifically at figure 3). However, the primary competitor algorithm is IPS which the proposed algorithm handily beats in three of the four problems. In the fourth problem, it is unclear that the competitor algorithm is winning, and could in fact be better only due to chance given the size of the respective error bars. It is worth noting that, because the parameter settings were tuned only for 50 trajectories, it is important to primarily assess performance based only on that point. It is likely that, given more trajectories to learn from, each algorithm would have chosen a smaller stepsize and effectively performed similarly.

Additional comments (did not affect score)

I would be interested in seeing the scalability of this approach empirically. Given the additional parameterized function to learn, I am unsure if this method would scale reasonably to much larger problems. However, I recognize that the scalability question is largely outside the scope of this paper.

**Experience Assessment:**

I have published one or two papers in this area.

**Review Assessment: Checking Correctness Of Derivations And Theory:**

I carefully checked the derivations and theory.

**Review Assessment: Checking Correctness Of Experiments:**

I carefully checked the experiments.

**Review Assessment: Thoroughness In Paper Reading:**

I read the paper at least twice and used my best judgement in assessing the paper.

---

> ### Author Response · Authors · 2019-11-15
> **Response to Review #2**
>
> Please find our detailed response in the following.
>
>
> Re "I am concerned with the consistency argument made in section 4.3. It appears from equation (10) and the final statement of theorem 4.1 that it is necessary to have two independent samples of x' given a single state x. In the general case, without access to the environment model, it is not possible to obtain two samples of x'. If the environment is continuing, then the probability of returning to state x to obtain a second sample is 0. Am I misunderstanding the requirements specified by the objective function in equation (10)?":
>
> The reviewer might have confused our independence condition with the double-sample condition in the RL literature (e.g., residual gradient).  The latter requires double samples of the form, $(s,a,s_1’)$ and $(s,a,s_2’)$, where $s_1’$ and $s_2’$ are independent next-states of $(s,a)$.  In contrast, our independence condition refers to the pair of samples, $(s_1,a_1,s_1’)$ and $(s_2,a_2,s_2’)$, where $(s_1,a_1)$ and $(s_2,a_2)$ are independent.
> Our independence condition can be met in several ways.  For example, if $(s_1,a_1)$ and $(s_2,a_2)$ are from two trajectories, they are independent automatically.  As another example, if they are from steps in the same trajectory that are far away from each other, then they are nearly independent under certain mixing assumptions (e.g., Assumption 2 in https://doi.org/10.1007/s10994-007-5038-2).
> We modified Theorem 4.1 slightly to make it clearer.
> We will add a discussion to clarify this condition in the final version of our paper.
>
>
>
> Re "An additional concern with the consistency argument is that it appears to assume that the approximator for d_w can achieve 0 error according to the MMD. If d_pi is not representable by the approximator, which could reasonably be the case in more challenging domains, it is unclear from this analysis or empirical results what the behavior of the system will be. It is difficult to assess how large of an assumption this is for consistency claim; for difficult problems will d_pi never be representable, or is this a fairly low concern?":
>
> The consistency result is a necessary (and important) verification that our objective function is fundamentally sound.  It implies that our approach will find the correct weights \emph{in the limit} --- with enough samples and using sufficiently rich function approximators.
> In practice, the reviewer is right that the performance of our approach depends on the representation power of the approximation function class.  The impact of such "approximation error" also exists in other RL algorithms (such as using deep Q-network when solving the Bellman equation) and in machine learning in general, and is not specific to this work.
> Minor point: our estimator aims to approximate $w(s,a)$, not $d_\pi(s,a)$.
>
>
>
> Re "The empirical results look to have high enough variance in the final outcomes that it is difficult to consistently assess the performance of each algorithm (looking specifically at figure 3). However, the primary competitor algorithm is IPS which the proposed algorithm handily beats in three of the four problems. In the fourth problem, it is unclear that the competitor algorithm is winning, and could in fact be better only due to chance given the size of the respective error bars.":
>
> In the revised version, we obtain slightly better results for our method with a retuning of hyperparameter.  As shown in Figs 2&3, it now outperforms 3 out of 4 tasks and performs comparably in the last one.
> Regarding variance: this work is inspired by Liu et al. 2018 to reduce variance, compared to standard methods that importance-reweight the entire trajectory.  But in general off-policy estimation is a very challenging problem, and state-of-the-art methods in the published literature still have relatively high variance. It is an important direction for future research.
>
>
>
> Re " It is worth noting that, because the parameter settings were tuned only for 50 trajectories, it is important to primarily assess performance based only on that point. It is likely that, given more trajectories to learn from, each algorithm would have chosen a smaller stepsize and effectively performed similarly.":
>
> Thank you for the suggestion which is a fair comment.  In the experiments, we set up the hyper-parameter this way in order to make sure our method (and other competitors) are not sensitive to hyper-parameters. This is important when applying such algorithms in practice.
>
>
>
> Re "I would be interested in seeing the scalability of this approach empirically.":
>
> We agree with the reviewer that scalability is important but outside the scope of the present work.  Here, our primary focus is to develop a new estimator, and to evaluate its usefulness in benchmarks.  We are indeed interested in investigating scalability as one of the future directions.

---

> > ### Comment · AnonReviewer2 · 2019-11-15
> > **Reply**
> >
> > Thank you for including an explanation of how $x$ and $\bar x$ are sampled. My concern was indeed about the double sampling issue present in RG methods, but now that the paper explains that two independent trajectories are used this concern is relieved.
> >
> >
> > "Minor point: our estimator aims to approximate $w(s, a)$, not $d_\pi(s, a)$."
> >
> > I see, I believe I have found the underlying source of my confusion. I had thought that the estimation of the state distribution was then being used as the importance re-weighting term in another estimator for the sum of rewards. In this case my concern about the consistency due to model class is ill-founded because there is not a compounding effect of model bias, only the omnipresent model bias common to most machine learning domains.
> >
> > Minor comment: why use $\rho$ instead of the more standard $G$ or $R$ for the return of the trajectory? I find that $\rho$ is overloaded as it is in the off-policy literature, and confused me greatly on rereading this work today after not having looked at the paper for a couple of weeks.
> >
> >
> > For the empirical results, my comments on variance were actually less concerned with whether the proposed method decreases the variance that is common in off-policy work, and more focused on whether the presented results were statistically significant. The high variance shown by the proposed method makes me question the statistical significance. I am okay with the fact that the proposed method appears to have higher estimator variance over independent runs than the competitors, but I'm curious how that plays a role in the significance of the results.
> >
> >
> > I'm actually particularly concerned, now, about the new results on the fourth problem. How did new meta-parameter tuning allow for better performance of the proposed method? Was this new meta-parameter tuning done for all competitors as well, and was it fair to those competitors? By re-tuning the proposed method, this could inject a strong bias in the results.

---

> > > ### Author Response · Authors · 2019-11-15
> > > **Our Response**
> > >
> > > Thanks for the quick reply.  We are pleased that our earlier response was helpful.
> > >
> > >
> > > Re "Minor comment: why use $\rho$  instead of the more standard $G$ or $R$  for the return of the trajectory?"
> > >
> > > Thanks for the suggestion.  We are open to changing notation to more common choices in the literature.
> > >
> > >
> > >
> > > Re "For the empirical results, my comments on variance were actually less concerned with whether the proposed method decreases the variance that is common in off-policy work, and more focused on whether the presented results were statistically significant. The high variance shown by the proposed method makes me question the statistical significance. I am okay with the fact that the proposed method appears to have higher estimator variance over independent runs than the competitors, but I'm curious how that plays a role in the significance of the results.":
> > >
> > > One way to address this question is to run the experiments for many times to reduce the standard error. We can do this if it becomes a point of concern. However, please note that our results’ significance is strengthened by observing that our method compares favorably to IPS in most cases, and comparably in the last one.
> > >
> > >
> > >
> > > Re "I'm actually particularly concerned, now, about the new results on the fourth problem. How did new meta-parameter tuning allow for better performance of the proposed method? Was this new meta-parameter tuning done for all competitors as well, and was it fair to those competitors? By re-tuning the proposed method, this could inject a strong bias in the results.":
> > >
> > > Regarding the hyper-parameter re-tuning, please note that not all methods share the same set of hyper-parameters. Therefore, simultaneously changing a hyper-parameter for all the methods might not be feasible because it may or may not appear in different estimators. For the 4th problem, we particularly changed the network architecture while keeping it fixed to a 3-layer network. Please note that both IPS and our method uses a neural network in their estimators. We re-tuned the network architecture for both during the rebuttal. We did not see any improvement for IPS but our method improved by re-tuning the network architecture. We should emphasize here that we tested every architecture that we used for our method on IPS as well. Therefore, in that sense, we believe that this re-tuning has been fair to our competitors.

---

### Official Review · AnonReviewer1 · 2019-10-22
**Official Blind Review #1**

**Rating:** 6

**Review:**

Summary :

This paper proposes an approach for long horizon off-policy evaluation, for the case where the behaviour policy may not be known. The main idea of the paper is to propose an estimator for off-policy evaluation, that computes the ratio of the stationary state distributions (similar to recent previous work Liu et al.,) but when the behaviour policy may be unknown. This is an important contribution since in most practical off-policy settings, the behaviour policy used to collect the data may not necessarily be known.


Comments and Questions :

	- This is an important and novel contribution where the authors propose an approach for OPE that comes down to solving a fixed point operator. This is simlar to solving for fixed point Bellman operators, but where their problem can be formulated as a backward recursive problem.
	- The operator defined as in equation 3 is in terms of the backward view of time - this allows the operator to capture the visitation from the previous s,a to the current s. This is the backward flow operator with which a fixed point equation can be described. Although the authors note that similar operators have appeared in the literature before - their main contribution is in terms of using such operators for the OPE problem, which seems novel to me and is an interesting approach with potential benefits as demonstrated in this paper.
	- The core idea comes from equation 9 which tries to minimize the discrepancy between the empirical distribution and the stationary distribution. This can be formulated as an optimization problem, and the paper uses blackbox estimators, as described in section 4.2 for solving this problem.
	- The next interesting part of the paper comes from solving the optimization problem in equation 9 with Maximum Mean Discrepancy (MMD) - this is a popular approach that has recently become well-known, and the authors make use of it minimize the differences between the empirical and the stationary state distribution.
	- Section 4.2 appears a bit confusiing to me with some details missing - it would perhaps be useful if the authors could include more details for 4.2, especially with some explanations of how they arrived at the expression in equation 10. This would also make the paper more self-contained, for readers in the RL community perhaps not so well-read with literature on MMD.  Appendix C contains the detailed derivation, but more intuitive explanations might be useful for the context of the paper.
	- The proposed black box estimator seems quite useful as demonstrated in figures 2 and 3. Although the authors evaluate their approach of few simple domains - it would have been useful if there were more extensive experiments performed for OPE problems. This would be useful since from fig 2, it appears that the proposed method only outperforms in 3 out of 4 evaluated problems.
	- For experiments, it would also be useful to demonstrate the significance of not knowing the behaviour policy and what are the usefulness of it. The paper is motived in terms of unknown behaviour policies that generated the data - so few experiments that explicitly shows the benefit of it would perhaps strengthen the paper more.
	- I am curious to know more about the bias-variance trade-off of the proposed OPE estimator as well. Ie, does the proposed method introduce any bias, or has significance in terms of lower variance for the long horizon problem? Experimentally, would it be possible to demosntrate whether the approach has lower variance compared to existing baselines?

Score :

- Overall, I think the paper has useful contributions. It is a well written paper, but some additonal details in section 4.2 might be useful, especially on the appriach with MMD. Experimentally, I think there are some experiments missing and doing those can significantly strengthen the paper as well. The proposed method seems simple and elegant, and I would recommend for a weak acceptance of the paper.


**Experience Assessment:**

I have published one or two papers in this area.

**Review Assessment: Checking Correctness Of Derivations And Theory:**

I assessed the sensibility of the derivations and theory.

**Review Assessment: Checking Correctness Of Experiments:**

I assessed the sensibility of the experiments.

**Review Assessment: Thoroughness In Paper Reading:**

I read the paper at least twice and used my best judgement in assessing the paper.

---

> ### Author Response · Authors · 2019-11-15
> **Response to Review #1**
>
> Please find our detailed response in the following.
>
> Re "Section 4.2 appears a bit confusing to me":
>
> We edited Section 4.2 to make it more self-contained. In particular, we added a brief discussion on RKHS and how it is connected to the MMD approach. We also added an intuitive explanation on how MMD could be understood in terms of GANs and added few references for interested readers.
> We added more details to Appendix C to spell out several steps in the proof explicitly, and will be happy to add further details if the reviewer suggests so.
>
>
> Re "The proposed black box estimator seems quite useful":
>
> We are currently running new experiments on more problems and will report them in the final version of our paper. In Figs 2 & 3 of the initial submission, our method outperformed other methods in 3 out of 4 problems. We re-tuned the hyperparameters for the 4th problem and now our method is comparable to IPS according to the updated Fig 3.
>
>
> Re "For experiments, it would also be useful to demonstrate the significance of not knowing the behaviour policy":
>
> It is important to note that we do *not* claim the usefulness of not knowing the behavior policy, as the reviewer seems to suggest.  Instead, the claim is that our approach has the benefit of not requiring that knowledge, thus making the new estimator more broadly applicable than previous work like Liu et al. 2018.
> Furthermore, a greater assumption our approach removes from Liu et al. 2018 is that transitions be drawn from the stationary distribution $d_{\pi_0}$ of the behavior policy (please see Introduction).  In the revised version, we also included a simple yet informative numerical example in Appendix F, which demonstrates that this assumption is critical to the Liu et al. 2018 approach.  In particular, when the behavior trajectories are short (so are far from being mixed), the Liu et al. 2018 estimator is biased and converges to an incorrect estimate.
>
>
> Re " I am curious to know more about the bias-variance trade-off":
>
> We included in Appendix F a comparison with DualDICE and IPS on the simple but informative ModelWin problem.  The task is to estimate the infinite-horizon average reward, using transitions from short trajectories.
> Our method has lower bias.  DualDICE’s bias is introduced by using $\gamma<1$ to approximate infinite-horizon estimation target.  IPS’s bias is due to the violation of the assumption that transitions are drawn from the stationary distribution of the behavior policy. Overall, please note that because of the consistency of our estimator, the bias of our approach converges to 0 asymptotically.
> Our method has slightly higher variance.  DualDICE has lower variance thanks to $\gamma<1$, which controls the effective horizon (on the order of $1/(1-\gamma)$).
> We are also running more experiments on larger-scale benchmarks and will include them in the final version of our paper.

---

### Official Review · AnonReviewer3 · 2019-10-23
**Official Blind Review #3**

**Rating:** 6

**Review:**

This paper proposes a new algorithm to the off-policy evaluation problem in reinforcement learning, based on stationary state visitation. The proposed algorithm does not need to know the behavior policy probabilities, which provide a broader application scenario compared with previous work. It is based on directly solving the fixed point problem of the resulting target stationary distribution, instead of solving a fixed point problem about the stationary ratios.

In general, this paper is looking at an interesting problem with a lot of increasing focus these days. I think this paper contributes great idea and good theoretical analysis. I like the idea of matching the resulting target distribution instead of minimizing a loss over the ratio. However, several unclear places in the current version hurt the presentation of results. I would like to see them get improved and will increase my score accordingly if so.

Detailed comments:

1) The algorithm part could be presented more clearly. So far I did not see where the empirical operator \hat{B} is formally defined. The word *empirical* is also confusing to me in "B is approximated by empirical data" because B is not an expectation, but an *integral* which has no *empirical* opposite of it. For the equations on top of page 5, shouldn't they be k[, ] about empirical operator instead of the expected operator since the RHS is already in sample-based form?

2) Related with the last one, B has an integral. To approximate the integral, we only have one sample from the transition probability actually, and the sample state is not uniformly distributed. It needs some explanation of why that would not cause a problem to approximate the integral.

3) The current loss function is invariant to the scale of w at all. Since the w is normalized, this is not a problem for the resulting estimator, ideally. However, that can be a numerical issue for float numbers. It's possible that the output from function approximator w goes to 0 or \infty. Both cases can lead to NaN of the output/function approximator update eventually. I'd like to hear if the author has met this problem in the experiment or not, and how can that be fixed.

4) I have to point out, as just a slight con of this paper, the technique used in this paper is not that much different from Liu et al 2018. Since it minimizes a loss function which is a supremum over an RKHS, and the resulting empirical loss also has a similar form. It's nice to see the author provide some details of making it work with mini-batch. These details are important for function approximator as NN.

Minor point:
 - On page 12 the equations after "We have by the definition of D_k", I did not follow the second step of the equations.

Suggestion:
This paper study the similar settings (behavior-agnostic OPE), using similar method (on the stationary distribution) came out several months ago: https://arxiv.org/pdf/1906.04733.pdf. I knew it's unfair to ask the author to compare it with a very recent prior/parallel work. However since they are in such a similar case, and they have code available, is it possible to directly compare with the result from their code? https://github.com/google-research/google-research/tree/master/dual_dice


======= After rebuttal =======
The author's feedback clarified some of my concerns in the initial review. After reading the author's feedback and other reviews, I think this paper has enough show contribution to the related work. Some of my previous concerns (point 2 and 3) seems true for many related works in this area in general. I partly agree that it is not very fair to ask this paper to fix them. The updated version also presents the theory section in a more clear way. So I'd like to raise my score. I've no problem with acceptance, but I won't fully heart argue for acceptance.




**Experience Assessment:**

I have published in this field for several years.

**Review Assessment: Checking Correctness Of Derivations And Theory:**

I carefully checked the derivations and theory.

**Review Assessment: Checking Correctness Of Experiments:**

I assessed the sensibility of the experiments.

**Review Assessment: Thoroughness In Paper Reading:**

I read the paper at least twice and used my best judgement in assessing the paper.

---

> ### Author Response · Authors · 2019-11-15
> **Response to Review #3**
>
> Please find our detailed response in the following.
>
> 1) Thank you for the great suggestion.  We have now included a formal definition of $\hat{\mathcal{B}}_\pi$ in Section 4.1, as well as pseudocode in Algorithm 1. Also, regarding "equations on top of page 5", thanks for pointing out this issue. In the initial submission, we were using the empirical and expected operator interchangeably in these equations. We have fixed the notation in the updated version.
>
> 2) While we have one sampled transition for each state, there are many states along multiple trajectories in the behavior dataset.  Furthermore, we use neural nets to represent the ratios $w(s,a)$, which allows \emph{generalization} across different states, even though a state may appear only once in the dataset (say, in continuous state problems).  We did not have any difficulty with this issue in the experiments.
> The reviewer is correct that, in order to have good approximation, $(s,a,r,s’)$ tuples in the dataset $\mathcal{D}$ should be widespread across the state-action space.  This is a common condition for batch RL to work well in general and is not specific to our algorithm.
> More generally, one-sample approximation of integrals is used in many successful algorithms in RL, such as fitted Q-iteration and deep Q-learning, etc.
>
> 3) Based on our observations and experiments, for a wide range of hyper-parameters we did not see numerical issues with the output of the function approximator, i.e., the output is not close to $0$ or $\infty$. However, to be safe, we clipped the output values of the function approximator. This could be done either by numpy.clip or clamp in PyTorch or clip_by_value function in TensorFlow.
>
> 4) We use similar optimization techniques as in Liu et al. 2018, but should emphasize that our main contributions are in the new objective function, not how to optimize it.  In particular, Our work aims to remove two significant assumptions of Liu et al., namely (1) knowledge of behavior policy; (2) more importantly, samples are drawn from the invariant distribution $d_{\pi_0}$.
> In order to do so, we had to find a different fixed-point equation that led to the general objective function (9), which enables consistent estimation of the ratios $w(s,a)$. It is also in contrast to the learning target of Liu et al., which is a function of states only: $w(s)$.
>
> Re "Minor point":
> We have updated that part with more details.  Please see the revised version.
>
> Re "Suggestion":
> As described in the paper, DualDICE only works for the discounted reward criterion ($\gamma<1$). In contrast, our work considers the more general undiscounted criterion (Appendix A).
> For the purpose of comparison, we set $\gamma$ to be close to $1$ in DualDICE, as an approximation of the undiscounted case.  Therefore, DualDICE estimates could be biased, but using $\gamma<1$ can potentially help reduce variance.
> Appendix F now has a comparison with DualDICE on ModelWin, with larger scale experiments being run at the moment.  Two $\gamma$ values are used: $0.9$ and $0.9999$.  As expected, DualDICE tends to have a higher bias but lower variance than our approach.  The effect gets intensified with a smaller $gamma$ value.

---

### Author Response · Authors · 2019-11-15
**Summary of Changes**

We thank the reviewers for carefully reading our paper and providing constructive comments. Here we give a summary of changes we have made in the manuscript, where the more important changes are highlighted in blue.

- Adding the pseudo-code of our algorithm (in Algorithm 1 Appendix E).
- Stating the definition of the empirical version of our backward-flow operator and fixing the equations corresponding to it (Section 4.1).
- Adding more background information about RKHS and MMD in Section 4.2 to make it more self-contained.
- Adding comparison with the DualDICE method from bias-variance point of view (Appendix F).
- Retuning our experiment for the Acrobot problem and updating the corresponding plot.
- Clarifying the consistency argument of our method.
- Moving the plots corresponding to the robustness of our approach to changing the behavior policy (Fig. 4 in the initial submission and Fig. 5 in the revised version) to Appendix G.

---

### Decision · Program_Chairs · 2019-12-19

**Decision:**

Accept (Poster)

**Comment:**

This paper addresses an important and relevant problem in reinforcement learning: learning from off-policy data, taking into account the offsets in the visitation distribution of states. This has the promise of lowering variance even with long horizon roll-outs. Existing methods have required access to the behavior policy (or have required data from the stationary distribution). The novel proposed approach instead uses an alternative method, based on the fixed point of the "backward flow" operator, to calculate the importance ratios required for policy evaluation in discrete and continuous environments.

In the initial version of the submission, several concerns were expressed regarding both the quality of the paper and clarity. The authors have updated the paper to address these concerns to the satisfaction of the reviewers, who are now unanimously in favor of acceptance.